# Involvement of Oxidative Stress and Nutrition in the Anatomy of Orofacial Pain

**DOI:** 10.3390/ijms241713128

**Published:** 2023-08-23

**Authors:** Marzia Gianò, Caterina Franco, Stefania Castrezzati, Rita Rezzani

**Affiliations:** 1Anatomy and Physiopathology Division, Department of Clinical and Experimental Sciences, University of Brescia, 25123 Brescia, Italy; marzia.giano@unibs.it (M.G.); c.franco@unibs.it (C.F.); stefania.castrezzati@unibs.it (S.C.); 2Interdipartimental University Center of Research “Adaption and Regeneration of Tissues and Organs (ARTO)”, University of Brescia, 25123 Brescia, Italy; 3Italian Society for the Study of Orofacial Pain (Società Italiana Studio Dolore Orofacciale—SISDO), 25123 Brescia, Italy

**Keywords:** orofacial pain, acute, chronic, oxidative stress, inflammatory soup, diet and nutrition

## Abstract

Pain is a very important problem of our existence, and the attempt to understand it is one the oldest challenges in the history of medicine. In this review, we summarize what has been known about pain, its pathophysiology, and neuronal transmission. We focus on orofacial pain and its classification and features, knowing that is sometimes purely subjective and not well defined. We consider the physiology of orofacial pain, evaluating the findings on the main neurotransmitters; in particular, we describe the roles of glutamate as approximately 30–80% of total peripheric neurons associated with the trigeminal ganglia are glutamatergic. Moreover, we describe the important role of oxidative stress and its association with inflammation in the etiogenesis and modulation of pain in orofacial regions. We also explore the warning and protective function of orofacial pain and the possible action of antioxidant molecules, such as melatonin, and the potential influence of nutrition and diet on its pathophysiology. Hopefully, this will provide a solid background for future studies that would allow better treatment of noxious stimuli and for opening new avenues in the management of pain.

## 1. Definition of Pain

“Halfway between the world of emotions and the realm of sensations, the history of pain, refers back to history of experience” [1,2].

Pain is a very important problem of our existence, and the attempt to understand it is one the oldest challenges in the history of medicine [2,3,4,5,6]

With time, the understanding of the origin of pain has improved; now it is known that its origin can be physical and psychosocial, but its persistence might be due to life circumstances, gender, and related factors [2,7].

In 1976–1977, the International Association for the Study of Pain (IASP) asked a diverse group of scientists (including experts in neurology, neurophysiology, dentistry, and so on) to define “what is pain”. Today, their definition is the most cited in the field of pain. The scientists defined pain as “an unpleasant sensory and emotional experience associated with actual or potential tissue damage or described in terms of such damage”. This revised definition was unanimously accepted by the IASP Council in 2020. The IASP’s definition seems to divide emotional and physiological pain, but as reported by Bourke (2013), there is no truth in this division [5]. Emotional and psychological pain are both valid “pains”, although only emotional pain is described as “painful”. In agreement with Descartes, the definition of pain divided between mind and body is alive and well and is used by more and more scientific researchers and many pharmaceutical industries [5].

The misunderstanding of pain leads to misdiagnosis and some very important problems in many fields and in clinical orthodontic diseases [8]. It is important to remember that where there is pain, there is a sense of urgency; “wait and see” is not an option that the patient willingly accepts, but there is a need to solve the problem. Dentists must be aware of a multitude of conditions that cause the symptom of pain and how patients may experience and express pain differently. It is particularly important for endodontists to recognize when pain has an endodontic origin and to distinguish it from pain resulting from other orofacial conditions. This is because the latter cannot be successfully managed with endodontic procedures. 

In 2020, the International Classification of Orofacial Pain (ICOP) introduced a classification of orofacial pain (OFP) to provide a correct nomenclature for healthcare providers and researchers [9]. Many scientists from several fields collaborated on reaching this goal, and this classification is now widely accepted and used by clinicians and researchers [9]. This classification covers seven main categories, including dentoalveolar and anatomically related tissues, muscle pain, temporomandibular joint (TMJ) pain, and neuropathic and idiopathic pain (Table 1).

This classification considers two different pains: *primary pain* that is not related to other diseases and *secondary pain* identified by other disorders such as inflammation, structural changes in the tissues, muscle spasms, or injury. Furthermore, the ICOP identified subcategories, which are reported in Figure 1.

After this subdivision, the classification of OFP better explained the categories of pain linked to dentoalveolar and other anatomical structures (Figure 2).

Endodontists have an important challenge in the application of the ICOP categories; they must identify and correctly manage endodontic pain [9]. When the endodontic origin has been confirmed, the specialists must prepare the treatment and monitor the outcome. If the pain does not have an endodontic origin, the diagnosis requires an interplay with patients and other colleagues. Figure 3 indicates the proposed use of the ICOP in endodontic special care. 

In conclusion, it is important to discuss improving and setting several strategies in relation to different pains; the International Classification of Orofacial Pain (IOCP) stresses this evidence since the current levels of scientific support for endodontic diagnoses are very low and not well defined [9]. One of these important concepts that can be used for the quality of life of patients is due to functional and psychosocial changes in diet and thereby nutrition [10]. 

The present review is an attempt to deepen the knowledge around pain from an anatomical and pathophysiological point of view. We focused on OFP, also considering the correlation among OFP, oxidative stress, and nutrition/diet to open new avenues in the management of noxious stimuli in orofacial regions and beyond.

## 2. The Pathophysiological Process of Pain

Normally, *pain* is felt when signals originating in thinly myelinated and/or unmyelinated nociceptive afferents reach a conscious brain; its purpose is to protect the human body from dangerous situations by associating them with unpleasant sensations. 

For a long time, three types of pain have been mechanistically known; they are nociceptive, inflammatory, and neuropathic pain. Nociceptive pain is the baseline defense mechanism that protects the human body from potential harm and is the physiological activation of neural pathways by several and different stimuli that are potentially damaging; inflammatory pain and neuropathic pain are characterized by the altered or aberrant function of the anatomical structures in the CNS [11,12].

Nowadays, many types of pain exist with several biological functions, and for this reason, other terms can also be useful to identify more specific situations in which people can feel pain. 

The main terms used to talk about pain are:Pain: it’s an unpleasant sensory and emotional experience associated with actual or potential tissue damage or described in terms of such damage, which is an individual, subjective experience [13].Allodynia: it’s a type of pain that is due to a stimulus which does not normally provoke pain: neurop-athy, inflammation, and certain headache states (embedded in the context of hyperesthesia) [13,14,15].Hyperalgesia: it’s a condition in which it is possible to identify an increased response to a stimulus which is normally painful: the stimulus and response modes are basically the same (embedded in the context of hyperesthesia) [13,14,15].Hyperpathia: it’s a term that indicates a painful syndrome characterized by an abnormally painful reaction to a stimulus, especially a repetitive stimulus, as well as an increased threshold. This is a typical form of neuropathic pain [13].Hypoalgesia: it’s a condition in which it is possible to recognize less pain in response to a normally painful stimulus. This is typical of neural damage; stimulus and response modes are the same but with a lowered response [13,16].Analgesia: it’s a condition characterized by an absence of pain in response to stimulation which would normally be painful. This is observed after complete axotomy or nerve block. Not unpleasant [13].Hyperpathia: it’s a term that indicates a painful syndrome characterized by an abnormally painful reaction to a stimulus, especially a repetitive stimulus, as well as an increased threshold. This may occur with allodynia, hyperesthesia, hyperalgesia, or dysesthesia, typical of neuropathic pain syndromes [13,17].Paresthesia: it’s a condition in which it is possible to identify an abnormal sensation, whether spontaneous or evoked. This is typical of neuropathic pain syndromes [13].

*Hypersensibility* is a feeling associated with injuries, burns, and infections that cause ongoing pain and tenderness [13]; this is, generally, an inflammatory pain. 

Pain in response to a normally painless stimulus is defined as *allodynia*, representing an accompanying feature of several pain syndromes. On the contrary, the feeling of excessive pain in response to a stimulus expected to be painful is also defined as *hyperalgesia* [14,15,16,17,18]. *Allodynia* and *hyperalgesia* are considered the most common forms of *hyperesthesia*, a cutaneous sensitivity increase [15].

Classically, the allodynia and hyperalgesia caused by everyday injuries have been explained by a putative increase in the responsiveness of nociceptor endings (peripheral sensitization) resulting from chemical inflammatory mediators released in the injured tissue. 

Over time, as outlined above, the view around the feeling of pain has changed, and it has been given increasing importance. Pain is a physiological homeostatic mechanism for preserving health and avoiding morbidity [19]. 

## 3. The Transmission of Pain 

Pain is a multidimensional emotional experience; it has two main components: sensations and emotions. Due to its complexity and the problems in the research methods, most studies reported transmission only in cortical regions and lacked evidence of the neural pathway mechanisms that underlie the changes in pain perceptions, emotions, and learning [13,20]. New experimental tools and techniques have been used to identify many neuronal pathways linked to sensations and emotions. 

### 3.1. The Anatomy of Pain and Related Neuronal Transmission

Nociceptors, responsible for identifying tissue injuries, are stimulated by three types of noxious stimuli: mechanical, thermal, and chemical. Chemical stimuli, such as serotonin (5-HT), histamine, potassium ions, acids, and acetylcholine bind to the receptors, inducing changes in membrane permeability, whereas mechanical stimuli act on receptor permeability to ions. Other chemical stimuli, like prostaglandins and substance P (SP), do not activate pain receptors but indirectly induce changes in the membrane permeability [21,22,23].

Pain receptors are not encapsulated free nerve endings; they are linked to pseudo-unipolar neurons (T-neurons) with cell bodies, for example, in the dorsal root ganglia (DRGs). It is known that DRGs show several cell types for different noxious stimuli, but these neurons can express the same proteins following the threshold and signal quality in a different way [24]. Furthermore, it was suggested that the sensations reach the CNS using a preferred type of neuron or due to the integrative activity of different neuronal classes. In 2019, the existence of DRG neuron subtypes was demonstrated, showing relevant genes with different functions such as voltage-gated potassium channels; these findings opened new scientific fields for studying the intrinsic physiological properties of DRG neuron subtypes [25]. Recently, the transcriptomic classes of human DRG neurons have been identified; of note, several types of human DRGs have similar transcriptomic features to DRGs in mice, but several transcriptomic classes showed different genes with no clear equivalent in other species, including humans [26]. Gross similarity between species is seen, for example, between neurofilaments and myelinated and unmyelinated fibers, but there are many genes that show specialization in each species [26]. Shiers and colleagues showed a quantitative difference in TRPV1 expression between humans and mice, even though it plays fundamental roles in physiological and pathological processes in most species [27]. Therefore, it is necessary to confirm the expression levels of the specific genes using complementary approaches to link the DRG classes to their neuronal functions [26].

Sensory nerve fibers are classified according to their conduction velocity and the stimuli that activate them, and they include small-diameter and medium- to large-diameter myelinated afferent fibers, as well as small-diameter unmyelinated afferent fibers [28]. Respectively, fibers Aδ are myelinated fibers with a diameter of 1–5 µm, a conduction speed of 2–20 m/s and they mainly respond to a mechanothermal and tactile stimulus from skin. Fibers C are unmyelinated fibers with a diameter of 0.02–1.5 µm, a conduction speed <2 m/s and they mainly respond to a polymodal stimulus carrying the transmission of nociceptive/tactile sensation. In the end, fibers Aβ are myelinated fibers with a diameter of 6–12 µm, a conduction speed of >2 m/s and they mainly respond to a tactile and pressure stimulus from skin.

Another consideration to be made concerns the modulation of the pain stimulus. This is an endogenous process that is thought to provide survival advantages. The studies mentioned here are quite recent, but they are important in defining how transmission occurs, confirming what was said earlier. Namely, they confirm that in the face of similar damage, the message that reaches the cortical level is not necessarily similar. What investigations have led us to understand is that an endogenous mechanism exists, and it can dissociate and modulate (enhancing or diminishing) the transmission of pain depending on the external conditions. There are several modalities available to our organism, not just one, and they include segmental inhibition, the endogenous opioid system, and the descending inhibitory system in the CNS, for example [26].

Moreover, in this case, because nociceptors can release peptides and neurotransmitters, such as SP, calcitonin-gene-related peptide, and ATP, from their peripheral terminals, they are able to activate bidirectional communication, facilitating the production of the inflammatory soup and giving rise to a situation known as neurogenic inflammation [28,29,30]. 

Considerable progress has been made toward understanding the neurological underpinnings of memory, learning, and long-standing pain, thereby permitting greater insight into common anatomical systems and neurochemical substrates. For example, the N-methyl-D-aspartate (NMDA) receptors play an important role in synaptic plasticity. During pain, these receptors are implicated in central and peripheral sensitization and in visceral pain [31]. NMDA receptor activation is involved not only in learning and memory but also in pathological conditions such as chronic/persistent pain [32]. Much evidence showed that structural changes at postsynaptic sites are linked to the creation of memories after learning [33], but at the same time, the reorganization of synapses, cells, and circuits at the brain level could intervene in the maintenance of chronic pain involving several neurotransmitter factors [34,35,36,37,38,39].

### 3.2. The Neurochemistry of Nociception

Our sensory system works by converting environmental stimuli into electrochemical signals, generating an action potential. This concept is true for all sensory systems, and nociception is particularly interesting. In this case, the stimulus that the body perceives as painful can have a different origin and nature [28], and for this reason, an individual primary sensory neuron in the “pain pathway” can be considered “special” if compared with other receptors. In fact, it can detect a wide range of stimuli. This means that the whole signal transduction system is special: on one side, different chemical or physical stimuli can activate a single receptor at the same time, leading to a response from the cells; on the other side, each nociceptor has the capability to modulate the signal it has picked up.

Moreover, once the signals are transduced by the primary afferent terminals, the receptor potential activates several voltage-gated ion channels critical to the generation of action potentials that convey nociceptor signals to synapses in the dorsal horn (DH) of the spinal cord [30]. The external pain stimulus and then the electrical potential result in a synapse that, in most cases, is a chemical synapse involving the release of neurotransmitters. 

Glutamate (Glu) is the predominant excitatory neurotransmitter in all nociceptors; this means that synapses with a second-order neuron can be activated by the acute release of Glu. Glu will then be recognized by specific postsynaptic receptors, such as the alpha-amino-3-hydroxy-5-methyl-4-isoxazolepropionic acid (AMPA) receptor, projecting to supraspinal sites through crossed ventrolateral tracts. Moreover, Glu uses two principal types of receptors: ionotropic and metabotropic Glu receptors; ionotropic Glu receptors have an ion channel that is directly activated upon glutamate binding, whereas metabotropic Glu receptors activate ion channels via coupling to G-protein signaling systems [40,41].

Many studies suggested that the inhibition of the fast excitatory effects of Glu is mediated by several substances [42], and the genetic depletion of some Glu receptors completely abolished the transmission [43].

## 4. Acute Pain, Chronic Pain, and Their Anatomical Localization

Several characteristics can be considered to classify pain. First and foremost are its duration and its location. 

The duration is the characteristic that allows a very linear distinction between two different forms of pain. There is *acute pain*, which implies a sudden onset but also requires a rapid resolution, and *chronic*, persistent *pain*, which does not resolve in the short term.

Generally, acute pain is considered a natural and useful sensation that warns the individual of possible injury and leads to behaviors that avoid further injury [29]. Chronic pain, on the contrary, can be defined as a painful feeling that lasts more than 3 months and may have some element of central sensitization. 

In chronic pain, cognitive and emotional factors have a critical influence on pain perception [29]. In these cases, it is not always possible to find a one-size-fits-all solution, because the root cause of chronic pain is not easy to identify. For this reason, its management is mainly focused on promoting rehabilitation and maximizing quality of life rather than achieving healing [44,45]. 

The categorization of pain, in fact, basically stems from patients’ reports: for them, it is easier to recognize pain by identifying which part of the body hurts. In this regard, a first distinction recognizes the damaged tissue and the cause of pain. Considering this distinction, two types of pain can be recognized: *somatic* (subdivided into the superficial somatic structures, the musculoskeletal structures, the structures of the supply system, and the special sensory organs) and *neurogenic* (derived from the nervous structures such as the brain and brainstem, the spinal cord, the peripheral nervous system, and the autonomic nervous system) [46]. Overall, neuropathic pain is, as mentioned above, a form of pain associated with alteration/damage/irritation/stress of the nerve components, both central and peripheral. Moreover, in most cases, neuropathic pain becomes chronic pain, and this leads to other “extra” and “external” clinical signs and symptoms, such as extreme allodynia, tissue erythema, temperature and trophic changes, and swelling, affecting the patient’s social life and therefore the patient’s quality of life [46].

The second feature that we can consider in order to classify the pain is its localization. The site of pain allows a classification to be outlined, and it is possible to recognize (a) head and face pain (indicated as OFP), (b) thoracic pain, (c) abdominal pain, and (d) extremity pain. In general, the definition of the localization of the pain is one of the easier steps that the patient can deal with. Moreover, it is useful to consider that there are parts of our body that are more “exposed” to pain because of the complexity of the structures that lie in these districts. This is the case of OFP, as we discussed previously and as we better explain in the next paragraphs [46]. 

## 5. Anatomical and Physiological Pathway of Orofacial Pain 

It is important to remember that the trigeminal nerve V (CN V) is responsible for pain in the orofacial region, which is composed of the oral cavity (teeth, gingiva, and oral mucosa), face, jaw bone, and TMJ [47]. The physiological pathway of OFP includes primary afferent neurons, pathological modification of the trigeminal ganglia, nociception of the brainstem neurons, and transmission to the brain [48]. 

The OFP pathway is shown in Figure 1. In detail, OFP arrives from the trigeminal nerve to the trigeminal ganglia; the trigeminal ganglia are similar to the DH of the spinal cord [19], and then the pain signal reaches the second-order neurons inside the brainstem. 

There are three trigeminal nuclei in the brainstem that project the inputs to the ventral posterior medial (VPM) thalamic nucleus and medial thalamic nuclei (e.g., medial dorsal nuclei); finally, the sensations reach the brain [49,50,51,52]. The first spinal nucleus of the trigeminal nerve is the pars oralis, and the second is the pars interpolaris; both are responsible for a tactile sensation in the orofacial area. The third nucleus of the trigeminal nerve is the pars caudalis or trigeminal nucleus caudalis (TNC) and transports the pain in the involved area [53,54]. Among different cortical regions in which the noxious stimuli arrive, five major areas are responsible for pain perception: the primary and secondary somatosensory cortices (respectively S1 and S2), ACC, IS, and prefrontal cortex (PFC). The ACC is an important area that is activated by different noxious or painful stimuli [55]. In addition to the ACC, the insular cortex is normally activated by different stimuli, including the unpleasantness of pain [56,57]. 

### 5.1. Classification of Orofacial Pain

The classification of OFP is divided into acute and chronic pain for any type of noxious stimuli and, by causes, it is classified into nociceptive, inflammatory, and neuropathic pain as reported above [58] (Figure 2).

The nociceptive stimuli arrive at the CNS, leading to pain response as a withdrawal reflex and as a vital physiological sensation [48,59]. Inflammatory pain is due to damaged tissues; once a tissue is injured, the mediators of inflammation are released and activate pain perception [48,60]. Neuropathic pain is due to alterations in the peripheral nervous system or CNS [61]. Table 2 describes the main differences between acute and chronic orofacial pain in terms of duration, causes, symptoms, and more.

Many studies suggested that peripheral nerve injury induces sprouting of myelinated Aβ primary afferent fibers determining the development of pain, but other results unequivocally demonstrated that this is not the case [65,66,67]. Although the sprouting of Aβ afferent fibers is present, it is very limited. It is the phenotypic change of unmyelinated C-fiber primary afferents that is responsible for the development of the sensory disturbance [67,68,69].

Moreover, Rotpenpian and Yakkaphan (2021) [48] indicated that there is another pain classification name called *nociplastic pain*. This is a not well-defined pain; this means that it is unclassified pain such as a persistent idiopathic dentoalveolar stimulus with a persistent tooth and alveolar bone pain, without clinical or radiological detectable examination [70]. However, the physiopathology of nociplastic pain is still not known, and other studies are needed to better specify this type of classification [48].

Last but not least, there is another point is of fundamental relevance when considering OFP: comorbidities that could be associated with patients’ paraoral habits. These can include bruxism (clenching and grinding the teeth), biting of oral soft tissues (cheek, lip, and tongue), biting foreign objects (snuff, fingernails, pens, and pencils, and so forth), tooth tapping, and smoking (cigarettes, cigars, and pipes). All these behaviors can be considered risk factors not only for the onset of general orofacial pain, but also for specific pathologies (such as temporomandibular joint dysfunction). Moreover, even if they are not the root cause, they could increase symptoms and worsen the prognosis [8].

Finally, the knowledge of OFP must be improved with other findings about the clinical classification of dental or non-odontogenic pain origin [48].

Data highlight how complex it is to manage OFP and how important it is to understand its etiopathogenesis to avoid overtreatment and, above all, to correctly manage the quality of life of patients. 

### 5.2. Physiology of Orofacial Pain and Neurotransmitters in Primary Sensory Neurons

OFP is transported by three main branches of the trigeminal nerve (CN V): ophthalmic, maxillary, and mandibular [51]; the main branches of the CN V are reported in Figure 1. 

Nociceptors showed many proteins and channels that transport pain following thermal, chemical, and mechanical stimuli [51]. The first pain receptor cloned was TRPV1 [71]; it is responsible for transmitting pain signals to the CNS by changes in the influx of calcium (Ca^2+^) and sodium (Na^+^) ions. Mechanoreceptors that evaluate blood pressure and tissue deformations are also involved in OFP caused by mechanical forces generated by the movement of dentinal fluid [51]. New theories will be better defined; for example, the *hydrodynamic theory* considers the causes of OFP by evaluating mechanical forces generated by the movement of dentinal fluid.

Further, many chemical substances induce nociceptive transmission of pain with tissue acidosis and inflammation or injury. Acid-sensing ion channel 3 and the transient receptor potential (TRP) ion channels are two main acid sensors involved in proton-induced hyperalgesic priming [51,72,73]. TRP ion channels are molecular sensors that are able to catch different types of stimuli such as mechanical stress and temperature, as well as small molecules including capsaicin and lipids such as phosphatidylinositol 4,5-bisphosphate [73]. Considering diversity and sequence similarities, mammalian TRP channels can be classified into six subfamilies: TRPC (canonical), TRPV, TRPM (melastatin), TRPP (polycystin), TRPML (mucolipin), and TRPA (ankyrin) [73]. Moreover, it is possible to distinguish them according to their relationship with calcium ions. Two members—TRPV5 and TRPV6—of the vanilloid subfamily, for example, are permeable to calcium with high selectivity; differently, other channels are weakly selective, but they can also be completely not calcium selective at all (this is the case for TRPM4 and TRPM5) [73]. Moreover, the TRPV1, TRPV2, TRPV3, and TRPV4 channels are known to be involved in heat sensation, and TRPV3 and TRPV4 are known to be involved in warm sensation, whereas transient receptor potential ankyrin 1 (TRPA1) and TRPM8 are known to participate in cool and cold sensations [73,74]. 

Another interesting point concerns the neurotransmitters present in the trigeminal sensory neurons and in the caudal nucleus of the trigeminal nerve, which, as mentioned, is involved in OFP transport [75]. The specificity of the identified neurotransmitters, neurotrophic factors, their receptors, and their main functions are summarized in Table 3. 

Several anatomical studies demonstrated that approximately 30–80% of all neurons of the trigeminal ganglia are glutamatergic [76,77]. So, we decided to better explain the findings about Glu functions. It is known that glutamatergic neurons play important roles in OFP. Clinical and experimental studies suggested that the expression levels of Glu receptors increased in the trigeminal ganglia following OFP after inferior alveolar nerve injury [78,79,80]. Moreover, the use of Glu antagonists is known to reduce nociceptive trigeminal responses in different OFP models. These results suggested that peripheric Glu activates afferent neurons via its receptors, inducing OFP [76,81]. 

Macrophages, mast cells, epithelial, dendritic cells, and odontoblasts can produce Glu, which is released into the extracellular space and transported by two neuronal Glu transporters and excitatory amino acid transporters (EAATs); in this way, Glu is able to activate phospholipase C (PLC) and protein kinase ε (PKCε), inducing the stimulation of TRP channels (TRPV1 and TRPA1) which mediate nociceptive inputs. The neuronal and non-neuronal release of Glu and the role of many cells in OFP are reported in Figure 3.

## 6. Orofacial Pain: Sex and Gender 

In recent years, the study of sex and gender differences has become increasingly popular, not only considering the general health and wealth of the population but also considering the pathological condition of patients. The fact that males and females experience pain differently is a concept that has always been inherent in the common imagination, but it has only been more recent research that has given scientific contextualization to these differences [82,83]. 

Evidence demonstrates that the female gender has a higher prevalence of OFP, including TMJ pain, primary headaches, and neuropathic conditions [84,85,86,87,88]. Moreover, women seek treatment more often than men, at a ratio of about 2:1. However, there is no complete consensus on whether these apparent differences are mainly due to biological, sociocultural, or psychological factors or to the neuronal networks between these [89] (Figure 4). It is important to remember that the possible differences between the genders must be considered during clinical diagnosis and treatment [87]. 

## 7. Orofacial Pain and Warning–Protective Actions

As reported above, pain and OFP have warning and protective functions, reducing the effects of the noxious stimuli that trigger some reflex responses from the body to the CNS [90,91]. 

OFP can be caused by dental infections (e.g., caries, pulpitis periodontal disease, obstructed eruption of teeth), neoplastic lesions (e.g., oral cancer, tongue cancer, tumors of maxilla and mandibula), post-traumatic lesions (e.g., dental inures, damage of temporomandibular joint), and craniofacial procedures (e.g., periodontal procedures, placement of implant) [92,93]. Moreover, pain in the face and neck regions can originate from other organs with innervations such as the facial nerve and, obviously, the trigeminal nerve [94,95]. These points determine the clinical multidisciplinary questions that require many specialists in several scientific fields. Furthermore, many studies suggested that it is very important to start pharmacotherapy early enough for improving the local and general conditions of patients [91]. 

The basic pharmacological methods for treating pain were defined long ago [96], but modern analgesics interact with the opioid receptors located in the brain, spinal cord, and peripheral tissues, inhibiting nociception [97]. These substances have many mechanisms of action, but these pathways, specifically, induce the inhibition of prostaglandin and cyclooxygenase (COX) synthesis and activity [98,99]. COX exists in two isoforms (COX-1 and COX-2); COX-2 is responsible for pro-inflammatory effects, whereas COX-1 has physiological functions [100]. If the analgesics induce COX-1 inhibition, they could determine adverse effects on several systems such as cardiovascular and gastrointestinal systems [91,101]; if the analgesics act on COX-2 expression, they have a much better safety profile than the other substances [102]. Furthermore, some analgesics can act on nuclear factor kappa (NF-kB) and inducible nitric oxide synthase (iNOS) expression, directly and indirectly inducing a reduction in inflammation and oxidative stress [103,104,105,106]. This will be better discussed in the next paragraphs.

Given the several mechanisms responsible for OFP, it is important to know the different causes of this pain to improve the quality of a patient’s life. Then, it is important to define and classify OFP by evaluating its practical aspects.

### Orofacial Pain Classification Based on Its Practical Aspects 

The multidimensionality of OFP has led specialists to create several divisions of pain stimuli based on the practical aspects of this phenomenon (formation and causes) [91]. 

Based on the formation of pain, nine mechanisms have been identified: provoked pain caused by the mechanical, electrical, thermal, and chemical stimuli [107]; prolonged provoked pain [108]; spontaneous pain due to the inflammation process; continuous pain that corresponds to chronic pain and involves some periods of remission [109]; nocturnal pain due to the horizontal position of the body and sometimes due the inflammation process [110]; throbbing pain linked to the heart rate and due to the inflammation process in the tooth area; fresh pain due to an incident in a tooth or an area adjacent to a tooth [111]; acute pain [110]; radiating pain [112]. These mechanisms are shown in Figure 5 [91]. 

Based on the causes of pain, several types of OFP have been identified, and they are reported in Figure 6 [91]. This division takes into account acute and chronic pain linked to periodontal procedures, post-extraction, or other practical procedures that could determine inflammation, neuronal symptoms, and a general disease of a patient. 

## 8. Orofacial Pain and Oxidative Stress

As described throughout this review, pain manifests in various forms in the body and is involved in different processes; therefore, its pathophysiology is not completely understood. The evidence indicates a complex network including nitrosative and oxidative stress, cation signaling, and the inflammatory response [113,114]. 

Oxidative stress is a physiological pathway; it is characterized by an excessive increase in the production of free radicals, including reactive oxygen species (ROS) (hydroxyl radical OH^−^, hydrogen peroxide H_2_O_2_, superoxide anions O_2_^−^) and reactive nitrogen species (RNS) (nitric oxide NO^−^ and its derivatives) that counteract the physiological capacity of antioxidants to scavenge them [115]. The imbalance between oxidative and antioxidative agents is implicated in the pathophysiology of many pain-related conditions such as cancer [116], treatment with chemotherapeutic agents [117], diabetes, endometriosis [118,119], vascular disease, and central/ peripheral nervous system diseases including chronic pain (nerve injury) [120,121]. The results of several studies show that ROS are involved in persistent pain (including neuropathic and inflammatory pain) [122,123,124]. In 2010, Viggiano and colleagues [120] showed an increase in O_2_^−^ production in a model of OFP and its involvement in inflammation-related pain transmission [120]. Moreover, high levels of these ions and their persistence in postsynaptic neurons would appear to be associated with hyperalgesia [120,125,126,127]. In 2013, Kallenborn-Gerhardt and colleagues studied and showed the implication of ROS in neuropathic pain (NP) [128]; ROS increase the excitability of nociceptive neurons through several mechanisms (e.g., activating NMDA receptors and inhibiting the proteins involved in the regulation of glutamatergic transmission), leading to the loss of Glu homeostasis [129,130,131,132,133]. ROS/RNS also decrease synaptic GABA release and lead to dysfunction/death in GABAergic neurons [133,134,135], reducing inhibitory transmission. These studies allow us to point out a real, though still not very clear, association between oxidative stress and pain.

This same correlation can also be observed specifically for OFP. Temporomandibular diseases are the most common chronic OFP conditions, including persistent pain and dysfunction of the TMJ. Many scientists have evaluated oxidative stress as one of the causes of TMJ diseases [136]. In a pilot study, Rodriguez et al. (2011) showed higher levels of oxidative stress biomarkers such as 8-hydroxydeoxyguanosine (8-OHdG) and malondialdehyde (MDA) in individuals with TMJ diseases; their expression is directly proportional to pain intensity [137,138]. However, there seems to be high variability in oxidative status in TMD, which could be due not only to the chronicity and the progression of the disease but also to other factors such as psychological stress [139]. 

A correlation between OFP and oxidative stress is also shown by the action of ROS on pain signaling through Ca^2+^-permeable TRP channels [132,140,141] involved in noxious sensation. TRP channels (TRPA1, TRPM2, TRPV1, and TRPV4) are highly expressed in neurons linked to nociception, including trigeminal ganglia neurons where they act as biosensors for environmental changes [142,143,144,145,146]. In combination, excessive Ca^2+^ entry and increased mitochondrial ROS levels are involved in sciatic nerve injury causing neuropathic pain [147]. For instance, in urinary bladder disorders and orthodontic pain [148], TRPA1 activation leads to oxidative stress, and ROS activation of TRPV1 was found to contribute to diabetic sensory neuropathy [143], TMJ diseases, and dental pain [146]. ROS/RNS can activate TRP channels through oxidative modification of amino acids and indirectly through second messengers [149].

### Correlation between Oxidative Stress and Neuroinflammation in Pain

Oxidative stress also plays an important role in the neuroinflammation of pain throughout the body, including OFP. These two processes seem to be linked by a series of mutual interactions that are still not fully understood. Neuroinflammation is a form of inflammation that develops in the central and peripheral nervous system, characterized by various changes in the nervous system such as increased vascular permeability, infiltration of immune cells, and production of numerous inflammatory factors [150]. 

ROS/RSN act as intermediates in signal transduction, activating transcription factors such as NF-kB. This, in turn, leads to the production of pro-inflammatory cytokines (e.g., TNF-α and mitogen-activated protein kinase (MAPK) pathway), thereby inducing the activation of several pathways involved in neuroinflammatory modulation and exacerbating pain conditions [149,150]. 

Many studies have shown a close relationship between oxidative stress and neuroinflammation in OFP [150]; the production of nitroxidative and oxidative species causes pro-inflammatory responses through toll-like receptors [151]. Accordingly, in several models of neuropathic pain, a reduction in ROS levels is related to a decrease in pro-inflammatory cytokines and an increase in anti-inflammatory cytokines [150]. A study performed by Sandoval et al. showed the interconnection between cyclin-dependent kinase 5 activation and ROS production by nicotinamide adenine dinucleotide phosphate (NADPH)-oxidase 1 (NOX1) and NOX2/NADPH oxidase complexes during inflammatory pain [144]. A vicious cycle is created between oxidative stress and neuroinflammation, compromising the anti-inflammatory and antioxidant mechanisms; ROS production is inhibited by transforming growth factor-β (TGF-β) and IL-10 through NOX2 inactivation, but at the same time, antioxidant molecules and NOX deletion stimulate the expression of anti-inflammatory cytokines [150]. The correlation between neuroinflammation and oxidative stress is summarized in Figure 7. 

The role of the NRLP3 inflammasome is not entirely clear, but some reports have suggested that oxidative stress through ROS might induce inflammasome activation, perhaps involving the TRPM2 channel [152,153] and the subsequent release of inflammatory cytokines [154]. The NLRP3 inflammasome appears to be activated directly by ROS, which are produced mainly by mitochondria but also through calcium flux as a result of the activation of TRPM2 by nitroxidative species [151].

It has been shown that pro-inflammatory cytokines such as IL-1β, COX-2, and TNF-α play an important role in the exacerbation of neuropathic pain [154], so inhibition of this inflammatory response could be a possible therapeutic way to reduce and relieve pain. Non-steroidal anti-inflammatory drugs (NSAIDs) are the most used group of drugs in orofacial treatments. These compounds act by inhibiting the synthesis of prostaglandin and are inhibitors of COXs (in a specific way for both isoforms, COX-1 and COX-2), which when active induce increased expression of cytokines and chemokines (TNF-α, IL-1, and IL-6) [91]. In addition, some NSAIDs inhibit the action of NF-kB and the expression of iNOS and reduce the release of ROS, decreasing oxidative stress [91]. These mechanisms are summarized in Figure 8.

## 9. Possible Treatment of Pain 

Evidence of the involvement of oxidative stress in the pathophysiology of pain and OFP has led researchers to study the possible use of antioxidant molecules to correct the imbalance in oxidative status by producing a beneficial effect; this therapeutic strategy is also used in other diseases in which oxidative stress is involved [155,156,157]. To date, few antioxidant molecules have been prescribed for the treatment of pain, but there are several studies on their actions [158]; one of these molecules could be melatonin, which we will discuss later. 

Many scientists have also identified the role of nutrition and diet in the treatment of pain and in OFP. There is much evidence showing that diet has an important role in OFP, and we will report it in one of the next paragraphs.

### 9.1. Antioxidant Molecules for Pain Treatment

As introduced above, there has been an attempt to use antioxidant molecules in therapies designed to modulate pain. For example, vitamins C and E, as well as vitamin D, melatonin, and curcumin, are all molecules with antioxidant and anti-inflammatory properties that have been shown to reduce ROS production and suppress inflammation by acting on several mechanisms, reducing NF-kB activation and/or inflammatory cytokine production; their action appears to reduce pain and thus exerts a beneficial effect in the treatment of endometriosis [159,160,161,162,163,164,165]. Valsecchi et al. [166] showed that genistein (a soy isoflavone) has a neuroprotective action in neuropathic pain due to its antioxidant and anti-inflammatory functions [167]. Genistein suppresses NO synthase, inducible NO synthase, and NF-kB expression and consequently increases antioxidant enzymes and reduces pro-inflammatory cytokines [166,167,168]. *Crocus sativus* L. (or saffron) is utilized for the treatment of several conditions. Its main components, crocin and safranal, seem to have antioxidant and anti-inflammatory properties. Based on this assumption, Erfanparast et al. (2015) studied the effects of crocin and safranal administration in formalin-induced orofacial pain in rats, showing a suppression of the local oxidative and inflammatory status [169]. Moreover, co-administration with morphine induced antinociception, reducing pain [169]. In vitro and in vivo models were used to study how *Erythrina* can manage pain; *Erythrina* is a member of the legume family and produces many secondary metabolites, in particular phenolic compounds (isoflavones and flavones) which, thanks to their antioxidant abilities, inhibit pro-oxidant molecules and inflammatory pathways such as MAPK and NF-kB [170]. The mechanism is not completely understood, and more preclinical studies are needed.

#### 9.1.1. Melatonin and the Possibility to Modulate Pain 

Melatonin is an endogenous indolamine released from the pineal gland and extra-pineal tissue, and it performs many functions including regulating circadian rhythms and antioxidant, anti-inflammatory, anti-apoptotic, and immune-modulating functions [171,172]. Recent in vivo and in vitro studies have demonstrated the efficacy of melatonin in relation to pain syndromes [173,174]. Chronic pain patients seem to have lower levels of melatonin in their blood and urine [171]. Moreover, serotonin and L-tryptophan, melatonin precursors, were present in lower levels in patients with fibromyalgia, indicating the critical role of melatonin in chronic pain syndromes [175,176,177]. The use of melatonin in several chronic OFP pathologies such as myofascial pain [178], headache disorders [178,179], fibromyalgia [180,181], and TMDs supports the analgesic ability of this indolamine [174].

#### 9.1.2. The Mechanism of Action of Melatonin in Pain

Melatonin acts through the MT1/MT2 melatonin receptors expressed on the cell membrane and through RZR/ROR orphan nuclear receptors [182]. Melatonin can modulate and facilitate GABAergic transmission, increasing GABA content and GABA receptor affinity and density [171]. Several experiments have shown that the antinociceptive action of melatonin is linked to calcium channels [183]. As noted, calcium channels play an important role in the development of oxidative stress and inflammation in OFP. Melatonin has the capacity to decrease neuronal free intracellular calcium levels, suppressing voltage-dependent Ca^2+^ channels (TRPV1 and TRPM2) in the cultured DRG neurons [173]. Additionally, melatonin seems to modulate the functions of potassium channels, as demonstrated by Hemati et al. (2021) [173]. 

Inflammation is upregulated by various mediators such as 5-lipoxygenase (5-LOX) and COX-2 (derived from arachidonic acid), which are involved in pain perception. Melatonin possesses strong anti-inflammatory functions; it diminishes the expression of 5-LOX and inhibits COX-2 activity, ameliorating nociceptive pain with an analgesic effect [183,184,185,186,187]. 

Mounting evidence has shown that the administration of melatonin can also inhibit pro-inflammatory cytokines, downregulate NF-kB expression, and consequently reduce the release of TNF-α. The administration of melatonin coupled with neurostimulation therapies seems to act synergically for chronic pain management [171]. In a clinical study, the intake of 10 mg of melatonin before sleep for 14 days in patients with headaches decreased the intensity and frequency of attacks [188]. Furthermore, melatonin restored the circadian rhythm, reduced pain-associated anxiety, and improved sleep quality, which are often compromised in pain conditions [183]. 

The multiple functions of melatonin are summarized in Figure 9. 

These properties and actions of melatonin need to be further studied and observed through other clinical trials. 

### 9.2. Orofacial Pain and Melatonin 

As mentioned above, very little is known about the effects of melatonin in OFP both in experimental conditions and in clinical trials, and the findings are sometimes contradictory. 

Huang et al. (2013), using a model of OFP in mice, showed that there is a downregulation of MT1 receptor expression in TNC during the early stage of the pain [189]. The downregulation does not change in the trigeminal ganglia, but the results suggested that the decreased MT1 expression in TNC attenuates the analgesic effects of melatonin. 

Recently, Scarabelot et al. (2016) showed that melatonin administration exerts an antihyperalgesic effect in an animal model of OFP due to the inflammation process [190]. The effect is linked to the mechanisms by which melatonin interferes with inflammatory and allogenic processes, as reported above. 

To our knowledge, there is only one clinical article reporting the improvement of chronic OFP with the administration of melatonin [191]. The authors of this article suggested that the improvement is linked only to the role of melatonin in reducing insomnia in patients. 

These findings indicate the importance of developing studies for better exploring how melatonin acts in acute and chronic OFP.

### 9.3. Orofacial Pain and Nutrition

A body of evidence suggests that OFP is often associated with an increased risk of malnutrition [192,193,194,195], but the literature in this regard is scarce, and there are no definitive conclusions. There are a few works concerning this point, such as the works of Durham et al. (2015) [10] and Nasri-Heir and Touger-Decker (2022) [196]. These authors indicated the impact of nutritional status on OFP and the potential role of diet in the management of this disease. These articles lacked definitive conclusions but underlined that antioxidant status may play an important role in OFP and that antioxidant strategies based on nutrition can have beneficial effects [197,198,199].

Knowledge suggests that the inflammatory origin of OFP has a role in the pathophysiology of the condition. As reported above, the mechanisms by which antioxidants protect biological systems from free radical damage comprise the direct scavenging of ROS by the sequestration of free catalytic metal ions, with the inhibition of NF-kB, the inhibition of the cyclooxygenase pathway, and a reduction in lipid peroxidation [200,201]. Moreover, the fatty acid profile of the diet impacts the inflammatory processes associated with OFP, and omega-3 fatty acids reduce inflammation, acting negatively on pro-inflammatory prostaglandins and leukotriene B4. 

The results in favor of nutrition’s potential role in the pathophysiology of OFP have been obtained in in vitro and experimental studies and not in clinical trials. Clinical trials are fundamental for better evaluating the importance of nutrition in the etiology of OFP since several patients have experienced an improvement with a controlled diet. The patients reported some problems with eating an adequate and varied diet, drug interactions, and/or weight changes. Without indications, these patients may become deficient in essential nutrients that may impact general health and well-being [10]. 

These considerations suggest that evaluations of a patient’s diet and nutritional status must be included in the management of OFP. In this regard, Durham et al. (2015) [10] demonstrated that simple approaches for non-dietetic health professionals exist and are useful for limiting nonintentional weight change, dietary problems, and diet quality. These indications must be used when giving patients dietary guidance.

## 10. Conclusions 

Pain can be considered one of the most debilitating human diseases; it compromises the quality of life of people from all over the world, representing an economic and social burden to society. 

In this review, we considered the localization of pain in the human body; the noxious receptors are predominantly ubiquitous, but they are not distributed with the same density in all regions of the body. There are parts of the human body that are more “exposed” to pain, like the orofacial regions. 

Despite the great advancement in medicine, the treatment of pain remains one of the major challenges to overcome. The involvement of different and ambiguous mechanisms and the discovery over time of new mechanisms involved have led to difficulties in pain management and are a possible reason for the failure to find a definitive treatment. 

To date, pain and OFP therapies are unsatisfactory, and many have adverse consequences; therefore, the development of more effective treatments is a major goal. As described, one possible therapy is the administration of melatonin, but this must be combined with other treatments. In addition, based on the above, another target may be TRP channels—TRPV1, TRPV2, TRPV3, and TRPV4 channels that are known to be involved in heat sensation; TRPV3 and TRPV4 that are involved in the transmission of a warm sensation; and TRPA1 and TRPM8 that are known to be involved in cool and cold sensations. 

The discovery of the involvement of oxidative stress and inflammation is an important starting point; furthermore, we report that nutrition and diet have a considerable role in OFP. 

In this review, the authors underlined and stressed the importance of developing new therapies, showing pain in its complexity and some mechanisms, in order to improve the everyday life of individuals. 

## Data Availability

Not applicable.

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
