# Peer review of "Involvement of Oxidative Stress and Nutrition in the Anatomy of Orofacial Pain"

_ijms, 2023, doi:10.3390/ijms241713128_

Round 1

Reviewer 1 Report

Obviously, pain is a highly complex biological phenomenon and, therefore, a review on this subject must preferably be focused on some special aspects of pain to provide interesting, useful and preferably new information for the reader. This review encompasses many aspects of pain sensation and processing in the nervous system but it is not focused. The manuscript would benefit from putting the focus on the mechanisms of orofacial pain, instead of dealing with subjects of textbook knowledge of pain in general.

Additional comments

The section on the “Anatomy of pain and related neuronal transmission” is ambitious but lacks important information e.g., on the transmitters and receptors contained and expressed by primary sensory neurons. Moreover, recently several groups of DRG and trigeminal neurons have been identified by large-scale single-cell RNA sequencing (Usoskin et al., Nat Neurosci, 18, 145–153, 2015; Zheng et al., Neuron 103, 598–616, 2019; Nguyen et al, PLoS One e0185543,  2017;  Nguyen et al, Elife 10:e71752, 2021). These data should be mentioned since they provided a basis of a novel classification of spinal and trigeminal primary sensory neurons.

Although the sources of tables and figures taken from publications of other authors are indicated, these are essentially unmodified copies of the illustrations published by others. The resemblance of parts of the present manuscript to earlier publications (e.g., Bourne et al, http://dx.doi.org/10.1016/j.nec.2014.06.001; Rotpenpian and Yakkaphan, https://doi.org/10.1523/ENEURO.0535-20.2021)  in both content and illustrations is conspicuous.

For clarity, Table 1 and 3 should be reformatted. Further, the statements should be supported by relevant literature references. 

In contrast to the authors’ statement (line 170), pain receptors are typically not encapsulated.

The manuscript would benefit from a linguistic and stylistic revision.

I recommend the authors to undertake a thorough revision of the text by taking the above considerations into account.

The manuscript would benefit from a linguistic and stylistic revision.

Author Response

Reviewer 1

Obviously, pain is a highly complex biological phenomenon and, therefore, a review on this subject must preferably be focused on some special aspects of pain to provide interesting, useful and preferably new information for the reader. This review encompasses many aspects of pain sensation and processing in the nervous system but it is not focused. The manuscript would benefit from putting the focus on the mechanisms of orofacial pain, instead of dealing with subjects of textbook knowledge of pain in general.

We thank for the suggestions; we have made changes to the manuscript to make it more focused on orofacial pain:

4. Acute, chronic pain and its anatomical  localization

5.Anatomical and physiological pathway of orofacial pain

The section on the “Anatomy of pain and related neuronal transmission” is ambitious but lacks important information e.g., on the transmitters and receptors contained and expressed by primary sensory neurons. Moreover, recently several groups of DRG and trigeminal neurons have been identified by large-scale single-cell RNA sequencing (Usoskin et al., Nat Neurosci, 18, 145–153, 2015; Zheng et al., Neuron 103, 598–616, 2019; Nguyen et al, PLoS One e0185543,  2017;  Nguyen et al, Elife 10:e71752, 2021). These data should be mentioned since they provided a basis of a novel classification of spinal and trigeminal primary sensory neurons.

We thank the referee, and we added some parts on transmitters and receptors and we reported also the papers of the groups that the referee suggested:

3.1. “The anatomy and pain and related neuronal transmission”.

3.2. “The neurochemistry of nociception”.

5.2 “Physiology of orofacial pain and neurotransmitters in primary sensory neurons” and Table 5

Although the sources of tables and figures taken from publications of other authors are indicated, these are essentially unmodified copies of the illustrations published by others. The resemblance of parts of the present manuscript to earlier publications (e.g., Bourne et al, http://dx.doi.org/10.1016/j.nec.2014.06.001; Rotpenpian and Yakkaphan, https://doi.org/10.1523/ENEURO.0535-20.2021)  in both content and illustrations is conspicuous.

Thank you for your comments, we have removed some figures in relation to the other indications and inserted unmodified figures from the referenced publications with relevant license.

For clarity, Table 1 and 3 should be reformatted. Further, the statements should be supported by relevant literature references.

Thanks for the advice, we reformatted the tables and added more relevant references.

In contrast to the authors’ statement (line 170), pain receptors are typically not encapsulated.

Thanks for the annotation, it was a transcription error, in agreement with the suggestion we replaced with not encapsulated. 

The manuscript would benefit from a linguistic and stylistic revision

Thank you for the suggestion, the manuscript received language editing from the Studio Moretto Group Srl-Italian Headquarters as indicated in this revised version in the Fundings.

I recommend the authors to undertake a thorough revision of the text by taking the above considerations into account.

We hope to have revised the text properly and correctly and to have replied to the comments.

Reviewer 2 Report

-          Title: Pain: from history to orofacial pain, through oxidative stress and nutrition. Change the title.

-          In this manuscript there is too much information about pain which are known for decades. From perception, transmission, transduction, acute, chronic pain and so on.

-          I suggest the authors to shorten the text and take only orofacial pain and its characteristics and treatment to summarize (as oxidative stress and inflammation and its correlation with orofacial pain).

-          Manuscript which is submitted is too confusing and difficult for readers. Also, there are too much known facts which are not necessary for this kind of journal.

-          “In conclusion, the present review is an attempt to deepen understanding of the cul- 118

ture of pain from an anatomical point of view. Here, we review the literature on the main 119

aspects in which pain is involved and its pathophysiology also considering orofacial pain 120

(OFP) that is sometimes purely subjective and not well defined. Understanding the char- 121

acteristics of pain, that is classified as acute and chronic pain and its mechanisms is im- 122

portant to open up new avenues in the management of the pain of the noxious stimuli.”

 – I suggest to remove the first two word “in conclusion”, because this is the first part of the review.

Table 1 and 4 – Divide table with columns and rows. It will be more clear.

Author Response

Reviewer 2

Title: Pain: from history to orofacial pain, through oxidative stress and nutrition. Change the title.

Thank you for the suggestion, we changed the title to “Pain and Orofacial pain: from bench to bedside”

 In this manuscript there is too much information about pain which are known for decades. From perception, transmission, transduction, acute, chronic pain and so on.

I suggest the authors to shorten the text and take only orofacial pain and its characteristics and treatment to summarize (as oxidative stress and inflammation and its correlation with orofacial pain).

Thanks for the advice, we modified the text following the directions and focused on oxidative stress and inflammation in orofacial pain throughout the text.

We deleted some parts that are not important and we have more focused on the oxidative stress and inflammation reported in: 9.1 Correlation between oxidative stress and neuroinflammation in pain. Lines 711-743.

Manuscript which is submitted is too confusing and difficult for readers. Also, there are too much known facts which are not necessary for this kind of journal.

Thank you for the advice, we edited the manuscript and eliminated the parts related to known notions.

“In conclusion, the present review is an attempt to deepen understanding of the cul- 118

ture of pain from an anatomical point of view. Here, we review the literature on the main 119

aspects in which pain is involved and its pathophysiology also considering orofacial pain 120

(OFP) that is sometimes purely subjective and not well defined. Understanding the char- 121

acteristics of pain, that is classified as acute and chronic pain and its mechanisms is im- 122

portant to open up new avenues in the management of the pain of the noxious stimuli.”

 I suggest to remove the first two word “in conclusion”, because this is the first part of the review.

Thank you for the suggestion, we removed “In conclusion” and modulate this part.

Table 1 and 4 – Divide table with columns and rows. It will be more clear.

Thanks for the suggestions, we have made stylistic changes to the tables and added more references.

Round 2

Reviewer 1 Report

The authors have made an effort to improve their manuscript, but it is still not focused enough and contains many irrelevant and unnecessary information on a wide range of subjects involving CNS development, interesting but irrelevant historical aspects on the artistic expression of pain, and GABA, just to mention a few. For example, the faces shown from the work of Niccolò dell'Arca have little if any relevance to orofacial pain. It is worth mentioning that the artistic expression of pain in the arts would deserve a much thorough treatise. In many parts of the manuscript the phrasing is still confuse. 

Some issues which need corrections:

line 474-475: The statement that the trigeminal nerve transmits pain from the neck is incorrect. Moreover, some parts of the head are not innervated by the trigeminal nerve, but by cervical nerves.

Fig. 4. deals with mechanisms involved in the development of neuropathic pain including structural reorganization of spinal Aβ primary afferents. Although peripheral nerve injury induced re-arrangement of Aβ-afferent fibers has been suggested to result from sprouting of this class of myelinated afferents, subsequent studies unequivocally disclosed that this is not the case. Indeed, the apparent re-arrangement of Aβ-afferent fibers is explained by the phenotypic change of injured unmyelinated C-fiber primary afferents not by sprouting of myelinated spinal afferents (Eur J Neurosci 16:175–185, 2002; Neuroscience 116:621–627, 2003; Brain Res 964:218–227, 2003). 

lines 569-570: The first “pain receptor” cloned was the transient receptor potential vanilloid type 1 receptor, the capsaicin receptor (Nature, 389:816-824, 1997).

Throughout the manuscript the authors refer to TRP channels instead of referring to the specific TRP channels which are involved in the physiology/pathology of the function in question. 

line 796: “mascella” should read maxilla

The English of the manuscript needs further improvement.

In many parts of the manuscript the phrasing is still confuse. 

Some examples:

lines 200-202

lines 364-366

lines 454-456

lines 509-510

lines 547-549

lines 708-710

lines 758-760

line 977

line 998

lines 1009-1011

Author Response

The authors thanks the Rerefee for the comments and the indications.

We have modified the main text according to them as follow:

The authors have made an effort to improve their manuscript, but it is still not focused enough and contains many irrelevant and unnecessary information on a wide range of subjects involving CNS development, interesting but irrelevant historical aspects on the artistic expression of pain, and GABA, just to mention a few. For example, the faces shown from the work of Niccolò dell'Arca have little if any relevance to orofacial pain. It is worth mentioning that the artistic expression of pain in the arts would deserve a much thorough treatise. In many parts of the manuscript the phrasing is still confuse.

We thank the Referee for the considerations. We have remodulated the main text, removing some of the paragraphs that were mainly focused on the historical and cultural background and that have little if any relevance to orofacial pain. We changed the title of the first paragraph (line 32, paragraph 1) now indicated as “Definition of pain”.

Moreover, we have remodulated also the part of the texts in which we refer to the role of GABA in the neurotransmission of the painful stimulus (ending part, paragraph 3.2)

line 474-475: The statement that the trigeminal nerve transmits pain from the neck is incorrect. Moreover, some parts of the head are not innervated by the trigeminal nerve, but by cervical nerves.

We thank the Referee for the suggestion, we have remodulated the title trying to give a better definition of the orofacial region.

Regarding the trigeminal nerve, we mainly mentioned it although it is not the only nerve involved in innervation of the orofacial region because it is one of the nerve most frequently implicated in dental and endodontic pathologies.

Fig. 4. deals with mechanisms involved in the development of neuropathic pain including structural reorganization of spinal Aβ primary afferents. Although peripheral nerve injury induced re-arrangement of Aβ-afferent fibers has been suggested to result from sprouting of this class of myelinated afferents, subsequent studies unequivocally disclosed that this is not the case. Indeed, the apparent re-arrangement of Aβ-afferent fibers is explained by the phenotypic change of injured unmyelinated C-fiber primary afferents not by sprouting of myelinated spinal afferents (Eur J Neurosci 16:175–185, 2002; Neuroscience 116:621–627, 2003; Brain Res 964:218–227, 2003). 

We thank the Referee for the suggestion, we have removed the figure 4. Moreover, we tried to better delineate the role of the different nervous fibers in the transmission of the pain stimulus.

In particular, we focused our attention, referring to a larger bibliography, on the role of the unmyelinated C-fiber as You suggested.

Lines 569-570: The first “pain receptor” cloned was the transient receptor potential vanilloid type 1 receptor, the capsaicin receptor (Nature, 389:816-824, 1997).

We thank the Referee for the suggestions. We elaborated according to the recommended references on the early studies on TRPV1.

Throughout the manuscript the authors refer to TRP channels instead of referring to the specific TRP channels which are involved in the physiology/pathology of the function in question.

We thank the Referee for the suggestion. We have added a concluding sentence at the end of section 5.2 indicating in more detail the different types of TRPs studied.

Line 796: “mascella” should read maxilla

We thank the Referee for the suggestion, we corrected typos.

The English of the manuscript needs further improvement.

We thank the Referee for the suggestion, we went through the whole text and corrected the grammar and spelling errors.

Reviewer 2 Report

Dear authors,

Manuscript is improved. 
It is less confusing and easier for readers.

But, again there are a lot of information that are known and was written about them for a long time ago.

I think that you have to shorten your manuscript 

Read more times and delete all unnesecarry knon things.

Title is better, but Pain and orofacial pain is unpropriate

Conclusion should not have references. You

should conclude without other consideration and thoughts.

Also, there are a lot of typographic mistakes. You shoud improve english.

I suggest for editors to give you more time to reconsider your manuscript and send beck to us when you are sure in it.

Improve

Author Response

The authors thanks the Referee for the comments and the indications.

We are happy that the manuscript now appears more straightforward and readable.

We have modified the main text according to them as follow:

But, again there are a lot of information that are known and was written about them for a long time ago.

We have remodulated the main text, removing some of the paragraphs that were mainly focused on the historical and cultural background, using the first paragraph for only the definition of the concept of pain.

Title is better, but Pain and orofacial pain is unpropriate.

We thank the Referee for the suggestion, we have remodulated the title trying to avoid typos and repetitions. We hope that it is better and more focused on the aim of this review.

Moreover, we can submit another possible title for this article (only in this reply, but we can change if it is better): Orofacial pain:  Role of oxidative stress and nutrition.

Conclusion should not have references. You should conclude without other consideration and thoughts.

We thank the Referee for the suggestion, we have removed the redundant references in the final paragraph, and we rewrote the conclusion focusing only on what the review of the literature that we’ve done have summarized.

Also, there are a lot of typographic mistakes. You shoud improve english.

We thank the Referee for the suggestion, we went through the whole text and corrected the grammar and spelling errors.

I suggest for editors to give you more time to reconsider your manuscript and send beck to us when you are sure in it.

We would like to thank the Referee for his/her kind availability, but as the authors have worked hard, they hope the result will respect the objective of the review.

Round 3

Reviewer 1 Report

The authors have improved their manuscript by putting the focus on issues of orofacial pain and eliminating non-relevant information. The manuscript has certainly benefited from the revisions the authors have made. 

Lines 39-41: It should be mentioned that, in 2020, the IASP task force recommended that the definition of pain be revised to “An unpleasant sensory and emotional experience associated with, or resembling that associated with, actual or potential tissue damage".  This revised definition was unanimously accepted by the IASP Council in 2020. The authors may consider to refer to this fact.

Line 185: “mice do not express the gene for TRPV1” This is certainly not the case, since ample data have shown that TRPV1 receptors play fundamental roles in physiological and pathological processes in this species.

Author Response

Lines 39-41: It should be mentioned that, in 2020, the IASP task force recommended that the definition of pain be revised to “An unpleasant sensory and emotional experience associated with, or resembling that associated with, actual or potential tissue damage".  This revised definition was unanimously accepted by the IASP Council in 2020. The authors may consider referring to this fact.

We thank the Referee for the suggestion, we have better specified in the text the reference to the last definition of the IASP Council in 2020 (line 41).

Line 185: “mice do not express the gene for TRPV1” This is certainly not the case, since ample data have shown that TRPV1 receptors play fundamental roles in physiological and pathological processes in this species.

We thank the Referee for the suggestion, we have better indicate in the text the expression and the role of TRPV1 in the different species (line 185).